# Pursuit and Expression of Japanese Beauty Using Technology

**Naoko Tosa [1], Yunian Pang [1], Qin Yang [1] and Ryohei Nakatsu [2,*]**

[1]   Graduate School of Advanced Integrated Studies in Human Survivability, Kyoto University, Kyoto 606-8306, Japan; tosa.naoko.5c@kyoto-u.ac.jp (N.T.); pang.yunian.87r@st.kyoto-u.ac.jp (Y.P.); yang.qin.7e@kyoto-u.ac.jp (Q.Y.)
[2]   Design School, Kyoto University, Kyoto 606-8501, Japan
*   Correspondence: ryohei.nakatsu@design.kyoto-u.ac.jp

**Abstract:** We have been working on the creation of media art, utilizing technologies. In this paper, we have focused on media art created based on the visualization of fluid behaviors. This area is named "fluid dynamics" and there has been a variety of research in this area. However, most of the visualization results of the fluid dynamics show only stable fluid behaviors and a lack of unstable or, in other words, unpredictable behaviors that would be significant in the creation of art. To create various unstable or unpredictable fluid behaviors, we have developed and introduced several new methods to control fluid behaviors and created two media arts called "Sound of Ikebana" and "Genesis". Interestingly, people find and feel that there is Japanese beauty in these media arts, although they are created based on a natural phenomenon. This paper proposes the basic concept of media art based on the visualization of fluid dynamics and describes details of the methods that were developed by us to create unpredictable fluid dynamics-based phenomena. Also, we will discuss the relationship between Japanese beauty and physical phenomena represented by fluid dynamics.

**Keywords:** fluid dynamics; high-speed camera; media art; fluid art; Japanese beauty

---

## 1. Introduction

We have created media art in which new technologies play an essential role. Recently, we have been interested in the usage of a high-speed camera, through which we have found hidden beauty in various natural/physical phenomena that could be revealed. In particular, we have been interested in the fluid behaviors and have been trying to create media arts by capturing fluid behaviors using a high-speed camera. Based on this methodology, we have been trying to create new types of media art (Feng Chen and Tosa 2013).

This area is considered "fluid mechanics" or "fluid dynamics" and there has been a variety of research in this area (Munson et al. 2012; Bernard 2015). As some fluid motions look beautiful, there is another research area called "visualization of fluid motion" (Smits and Tee Tai 2012). One such beautiful fluid motion is the well-known "milk crown" (Krechetnikov and Homsy 2009). However, most visualization results show only stable fluid behaviors and a lack of unstable or, in other words, unpredictable behaviors that would be significant in the creation of art. Therefore, to realize various unstable or unpredictable fluid behaviors to create artworks, it is important to introduce several new methods.

In this paper we describe two methods that have been developed and introduced by us to create new media art. In one method, we used viscous fluids such as paints with various colors, to which we applied vibration to produce upward motion and shot their "jumping-up" behaviors. It was revealed that jumping-up paints create beautiful forms that change in a very short time. Such forms were shot

by a high-speed camera and then based on the editing of the obtained video, a new type of media art called "Sound of Ikebana" was created (Pang and Tosa 2015; Naoko Tosa et al. 2015).

We introduced a new method of letting color paints injected into fluid and dry ice bubbles interact to create beautiful forms of color paints, which led to the creation of a media art called "Genesis" (Naoko Tosa et al. 2017).

At the same time, we have received comments on these media arts from many Western people including art curators, art critics, etc. Interestingly, they feel that there is Japanese beauty in these artworks. Why do they feel Japanese beauty in the visualization of a natural/physical phenomenon? For this, based on our consideration, we have developed a hypothesis that one important factor of Japanese beauty is based on the extraction and expression of beauty hidden in natural/physical phenomena. As the relationship between our media arts and Japanese beauty is fundamental for the value of such arts, we will discuss what Japanese beauty is preceding the description of the media arts we have developed.

This paper consists of the following sections. In Section 2, a discussion on Japanese beauty is carried out and we make a hypothesis that one important factor of Japanese beauty is based on the visualization of hidden beauty in nature. In Section 3, the basic concept of the visualization of fluid dynamics as a method to create artworks is described. In Section 4, the detailed description of one type of media art creation method based on the fluid dynamics is described and the media art called "Sound of Ikebana" based on this method is described. In Section 5, the details of another type of art creation based on the fluid dynamics and also the created artwork called "Genesis" is described. Finally, in Section 6, we present the discussion and a conclusion is described.

## 2. Characteristics of Japanese Art

### 2.1. What Is Japanese Beauty?

What is the essence of Japanese beauty? As indicated by Bruno Taut and others, the harmony between humans and nature has always been emphasized and expressed in Japanese artworks and architecture (Taut 1958; Taut 1962). Trying to find out the root of such a basic concept, we reach the Chinese philosophers Lao-Tzu and Zhuangzi and their philosophy called "Taoism (Wong 2011)," in other words "Eastern Monism," which emphasizes the unification of humans and nature. Although Japanese beauty consists of various factors, based on this, it could be said that one factor of Japanese beauty is not beauty created by humans but beauty hidden in nature. Also, it could be said that one factor of Japanese beauty is what Japanese artists have tried to extract from nature based on their sensitivity and have expressed in the form of their artworks. This means that there is a close relationship between Japanese beauty and natural or physical phenomena. We noticed this based on our experiences described below.

We have focused on the creation of artworks based on the methodology of finding and extracting beauty hidden in natural/physical phenomena by using a high-speed camera. One of the authors, Naoko Tosa, was named as Japan's Cultural Envoy by the Agency of Cultural Affairs, the Japanese Government, in 2016 and exhibited her artworks in many cities all over the world. During such exhibitions, she received many responses from many people including art critics and art curators saying that "Naoko Tosa's artworks showing beauty hidden in nature express beauty that has not been noticed by Western people. Her artworks include the essence of Japanese sensitivity and consciousness".

It sounds a bit strange that Western people feel that there is Japanese beauty in artworks created based on natural/physical phenomena. Next, we will discuss this issue comparing Western and Eastern art history.

The creation of artworks based on beauty in nature is not an idea specific to Japan. This idea has been shared in many countries and cultures. In the West, since the Greek era, the idea that art is "imitation of nature" has long been accepted and this idea became the basis of the inventions of various art techniques such as perspective. However, since the late modern era, along with the invention of

the camera, this idea was gradually replaced by another idea that art is the "expression of humans' inner life" and this trend continues through art movements such as Impressionism, Cubism, Abstract Expressionism, and so on.

On the other hand, in the East, these theories have not been the mainstream in the art world and the basic concept of Eastern Monism that stresses the unification of humans and nature has been dominant. In contrast to Western artists, Eastern artists have neglected the concept of shadows and perspectives which play important roles in Western art. Having the idea of the unification of humans and nature deep in their minds and using their sensitivities, Eastern artists have created their artworks and also their own art world. In China, for example, monochrome ink paintings of landscapes have been popular. In such landscape paintings, based on the old Chinese philosophy of Taoism, Chinese artists tried to draw ideal landscapes—in other words, Arcadia.

As Japan used to continuously import Chinese cultures, Japanese art was deeply influenced by Chinese art. Then gradually merging this with the sensitivity of Japanese people, especially influenced by the isolation policy in the Edo era, Japanese artists began to create their own artworks without shadows, and being planar, exaggerated, etc.

As these Japanese artworks in the modern art era look very fresh to the West, who has denied the idea of "imitation of nature," in 19th century, the movement called Japonism occurred.

Consequently, we can interpret the impressions of Western people toward Naoko Tosa's artworks, when they say that her artworks express Japanese beauty, in the following way. As the concept of art in the West has changed from its original idea of "imitation of nature" to the modern and present one of the "expression of human's inner life or concept," Naoko Tosa's artworks, that are created based on capturing and extracting beauty in nature and that are a contrast to Western art, appealed to their sensitivity and made them feel that her artworks express Japanese beauty.

Based on this experience we can make a hypothesis that "One important factor comprising Japanese beauty is based is the extraction and expression of beauty in nature." For the extraction of such beauty, there could be several ways. One such method is based on the sensitivity or natural gifts of artists. Another method is based on the usage of technologies, which have been adopted by us.

In the next subsection, we will discuss several examples of Japanese artworks and artforms showing that one factor of Japanese beauty is based on the extraction of beauty hidden in nature and the creation of artworks containing such beauty.

## 2.2. Examples of Japanese Beauty in Japanese Art

In natural phenomena, such as water flow or wave forms, Japanese artists have found beauty and by expressing such beauty, they have created their artworks. One such artform is the well-known artworks by Katsushika Hokusai (Thompson and Wright 2015). Also, the specific expression of water flow, called "Korin wave," designed by Ogata Korin is very well known (Fujiura 2018). Such artworks are typical expressions of Japanese beauty and have been welcomed by Western artists, giving them strong impressions. Figure 1 shows Fugaku sanjurokkei Kanagawa oki Namiura (the Wave off Kanagawa, from 36 Views of Mountain Fuji), a print by Katsushika Hokusai (Clark 2017). Interestingly, the dynamic waveform expressed well resembles the wave form shot by a high-speed camera. Figure 2 illustrates the fluid form created by injecting air-gun bullets into fluid with color paints. It is interesting to know the resemblance between these two.

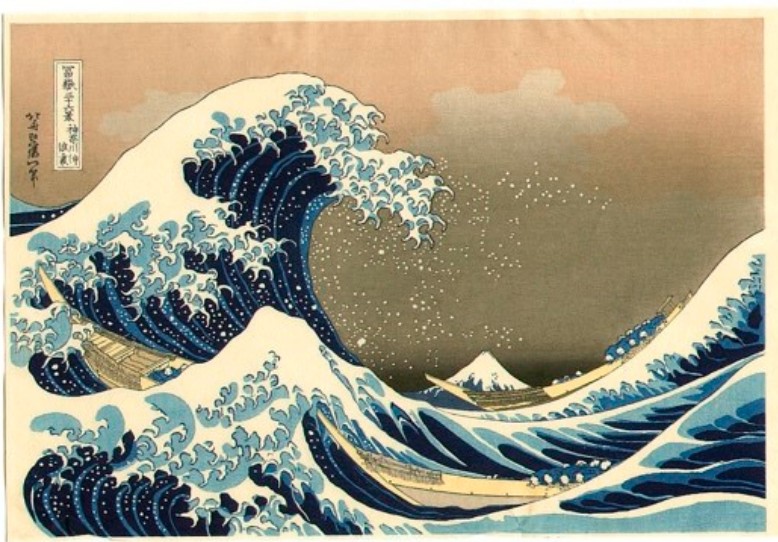

**Figure 1.** "Fugaku sanjurokkei Kanagawa oki namiura (the Wave off Kanagawa, from 36 Views of Mountain. Fuji" by Katsushika. (in public domain)

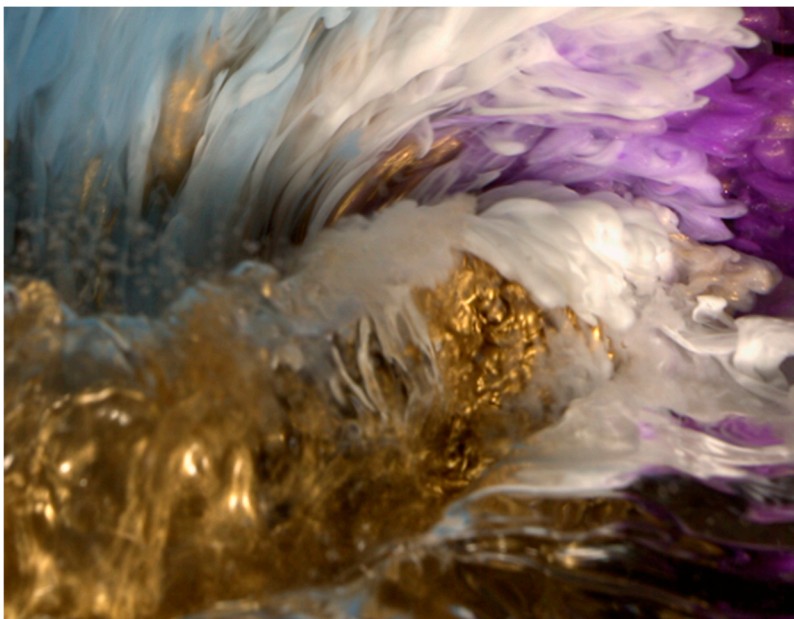

**Figure 2.** Fluid form captured by a high-speed camera.

Another example is a basic form of Japanese "Ikebana" (flower arrangement). The basic form of Ikebana has been considered an "asymmetric triangle" (Figure 3). We have succeeded in creating a similar form by letting color paints jump up by applying sound vibration and by shooting the jumped-up color paints by a high-speed camera, which is described later (Figure 4).

What produces this resemblance between the artworks and the form expressing Japanese beauty and natural/physical phenomena? Perhaps it is that great Japanese artists, such as Katsushika Hokusai, can find beauty hidden in natural/physical phenomena using their sensitivity and talent and can create artworks using the beauty found. For now, this remains a hypothesis, but we want to reveal this by continuing the creation of artworks based on beauty hidden in nature.

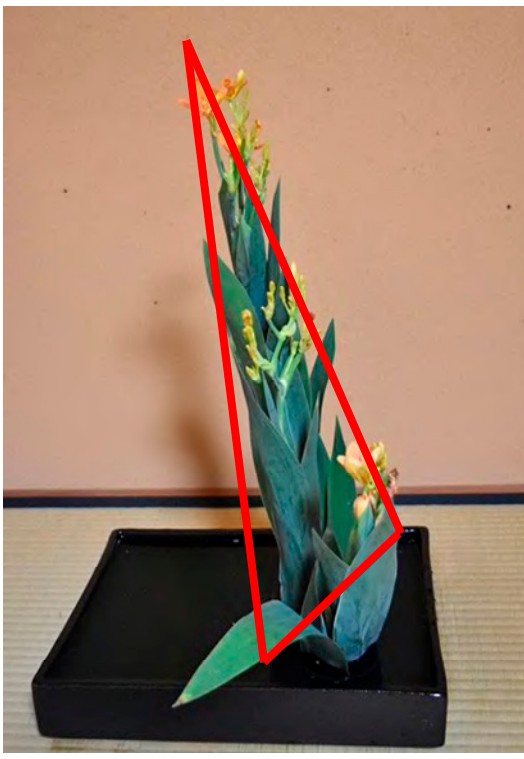

**Figure 3.** "Basic Form" of Ikebana. (in public domain).

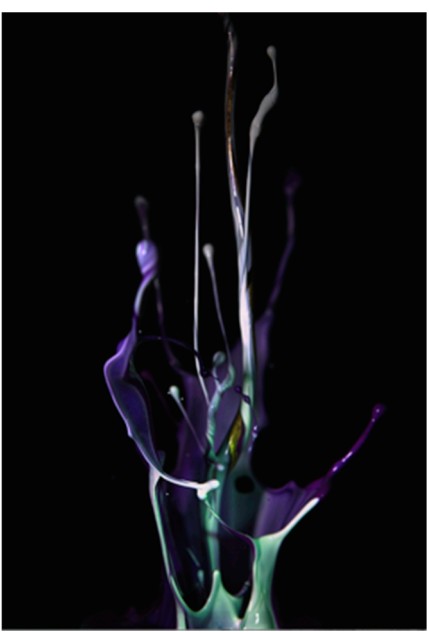

**Figure 4.** Fluid form created by sound vibration and captured by a high-speed camera.

## 3. Visualization of Fluid Dynamics as a Method to Create Media Art

Study of the behaviors of fluid has been a long-time research topic in physics and this area is called "fluid dynamics" (Munson et al. 2012; Bernard 2015). In physics, fluid dynamics is a sub-discipline of fluid mechanics that deals with fluid flow. It has several sub-disciplines itself, including aerodynamics (the study of air and other gases in motion) and hydrodynamics (the study of liquids in motion). Fluid dynamics has a wide range of applications including calculating forces and moments on aircraft, determining the mass flow rate of petroleum through pipelines, predicting weather patterns, understanding nebulae in interstellar space and modeling fission weapon detonation.

Determining how to explicitly show the behavior of fluid is another research area called "visualization of scientific phenomena" (Smits and Tee Tai 2012). Based on this visualization process, it became possible for people to watch the actual process of fluid behavior and it has been recognized that various beautiful fluid motions can be created depending on various conditions. As beauty is the fundamental element of art, utilizing fluid dynamics as a method to create artworks has been one of the key concepts of art creation. There are various artworks that utilize the concept of fluid dynamics. These approaches can be classified into two ways.

One approach is from a purely scientific side. Fluid motions, especially when there are obstacles in the pathway of the fluid, look beautiful and sometimes the visualized result of such fluid motion is considered art. Figure 5 shows the result of the visualization of stable flow called "laminar flow". As the ratio between inertia and viscosity, called "Reynolds number," increases, the laminar flow changes into unstable flow called "turbulence." In turbulence, frequently various types of vortex occur, some of which look beautiful. Figure 6 shows one example of such a vortex.

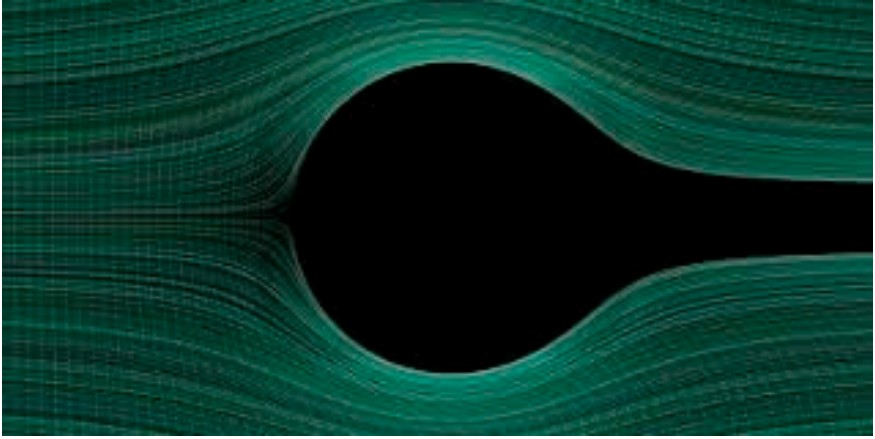

**Figure 5.** An example of laminar flow.

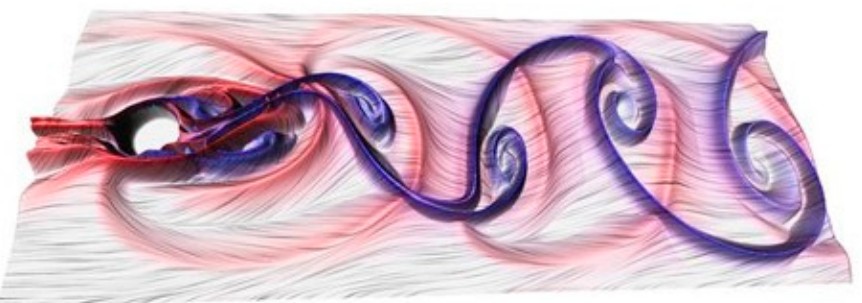

**Figure 6.** An example of a vortex.

Although various types of beautiful forms can be created based on such approaches, created forms based on such approaches are not considered pure art. The reason for this is that these phenomena or created forms are still based too much on physics and it is difficult to include an "intention of artists" in the form creation process. There is a clear distinction between physical phenomena and artworks and the border is how much intention of the artists to create artworks is involved in the created work. If there is no intention or the intention is too weak, the created forms are considered physical phenomena rather than artworks. In other words, forms created as physical phenomena are controlled by the laws of physics and there is little space for where something unexpected happens and this unexpectedness is a core part of artworks.

On the other hand, there is a different approach which is from an art basis. In this case, fluid usage is strongly controlled by the artists and unexpected phenomena or chance phenomena that

happen in the process of fluid usage are utilized by the artists to include something unexpected into their artworks. One representative of such art creation processes is "Action Painting" (Fleck et al. 2008) led by Jackson Pollock (Landau 2010). Action painting is a form of art creation in which, instead of drawing paintings using a paintbrush, artists throw or draw paints on a canvas. Basically, artists have intentions regarding what kind of paints they use and where on the canvas they throw or draw paints. Therefore, in addition to the intentions of the artists, a kind of contingency caused by thrown or drawn paints influences the final form of the created artwork. Figure 7 shows one of the representative artworks of Jackson Pollock. Although he is now highly evaluated and appreciated in the modern history of art, a problem with his artworks is that it is difficult to find natural beauty, and, therefore, in the beginning, this confused many people.

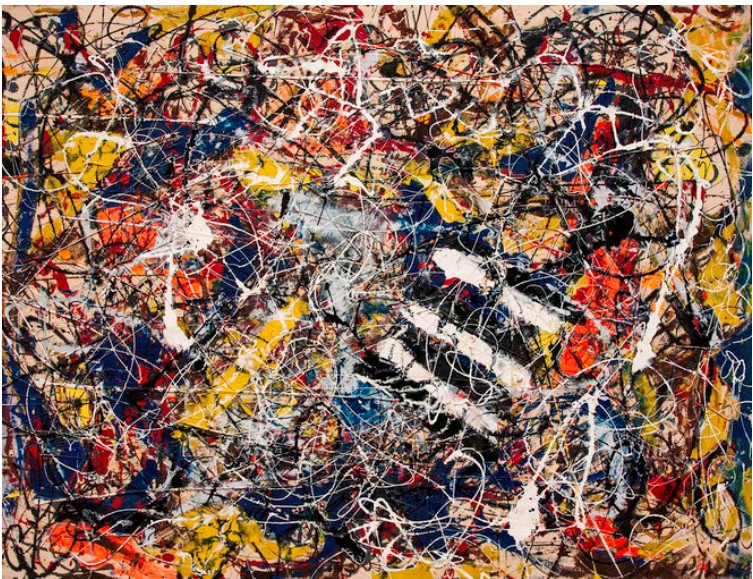

**Figure 7.** One of the Jackson Pollock's drawings (in public domain).

Based on the problems included in these two approaches, we think that there should be another way of new art creation somewhere in between these two approaches. We started from the former approach but tried to include more unexpectedness in created forms. In Sections 4 and 5, two methods to realize this are described.

## 4. Sound of Ikebana: An Example of Created Art

As one method to create artworks based on the visualization of fluid dynamics, we have developed a method to combine color ink fluid and sound vibration.

### 4.1. Sound Vibration System

It is well known that applying vibrations to liquids such as water creates movement in the liquid. For example, putting water on a drum and playing the drum creates a beautiful water splash form and this is frequently used as a performance. This is visible beauty based on a physical phenomenon. Inspired by this, we wanted to find invisible beauty included in this type of physical phenomenon. To realize this, we introduced a high-speed camera as key equipment and have developed a system to realize and shoot such physical phenomena called a "sound vibration system".

The sound vibration system is a new art creation method, which generates various changing shapes of materials ejected up by sound vibration (Yunian Pang et al. 2017). We used a high-speed camera with the rate of 2000 frames per second, and replayed it with 30 frames per second. This means we expand real time to 67 times. Then, the beautiful phenomenon hidden in nature is able to be seen by us directly.

The top-down view of the system is illustrated in Figure 8. First, we placed a rubber sheet over the top of a bass speaker and stretched the rubber to give it enough tension. Then, we fixed the rubber to make it stable. After that, we poured various fluid materials, with carefully controlled quantities and viscosities, onto the rubber. A rap-top computer was used to generate sound with various wave shapes and frequencies, and the generated sound was fed to the speaker. The vibration of the sound was then delivered to the rubber and to the color paints on it. The color paints were forced to jump from the rubber rapidly. A high-speed camera was used to record the changing shape and another computer connected to the camera recorded this. Also, to realize enough brightness for better quality of the shot video, we introduced two 300 W xenon lamps.

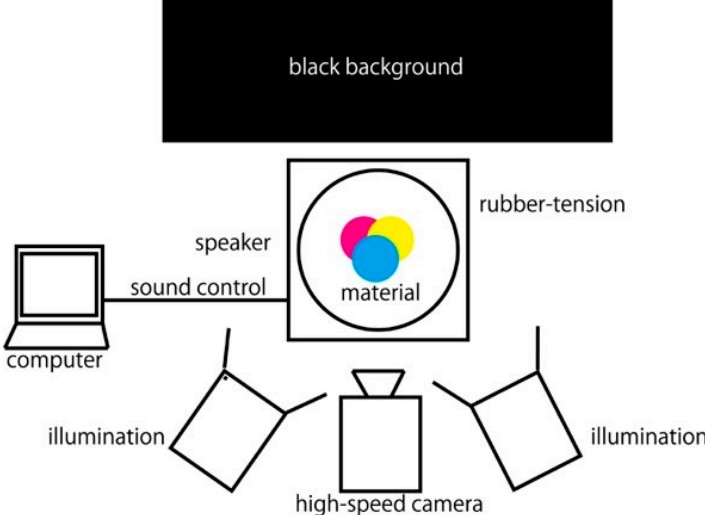

**Figure 8.** Top-down view of the sound vibration form system.

### 4.2. Sound of Ikebana: Created Art Based on SVF

By using the sound vibration system described above, we carried out various experiments by changing the type of sound, sound frequency, sound volume, liquid type, liquid viscosity, etc., and based on this, we created an artwork called "Sound of Ikebana." In this artwork, sound was used as an energy source which can eject color paint up above the speaker. Then a high-speed camera was used to capture the motion of the paints. By expanding the time of the phenomena, we can see the beautiful shape of the paint, which looks like "Ikebana," the Japanese flower arrangement. As was described in Section 2.2, it is interesting to see the similarity between various forms created by sound vibration and Ikebana, a typical traditional Japanese culture.

This artwork is a combination of the latest technology and the traditional Japanese flower arrangement culture. Sound of Ikebana consists of four short videos, each of which represents one of the four seasons in Japan. It uses specific colors to represent flowers in each season (Figure 9). By utilizing various types of color paints and liquids, we tried to express Japanese flowers in each season, such as plum and cherry in spring, cool water and morning glory in summer, red leaves in autumn, and snow and camellia in winter. Additionally, we tried to express various color variations such as prayerful colors of Buddhism, Japanese "Wabi" (austere beauty) and "Sabi" (elegant simplisity) colors, colors of delicious food, cute colors of "Cool Japan," gorgeous colors featuring the New Year season, etc.

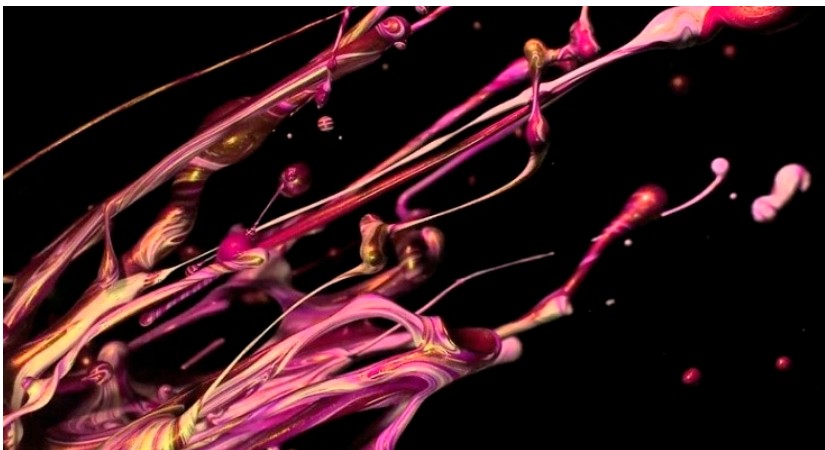

**Figure 9.** A scene from "Sound of Ikebana".

By watching these series of video artworks, the audience would have a feeling of wonder generated by the organic and mysterious figures of the liquid and also its unforeseeable movements. At the same time, the audience would feel the connection of the long history and traditional cultures in Asia.

To display artworks to many audiences in an effective way, a projection mapping has been frequently used. We carried out the projection mapping of Sound of Ikebana at Singapore ArtScience Museum in 2013. The moving images of Sound of Ikebana were projected on the wall of the lotus-like ArtScience Museum. The artwork became a part of the city night view, and the whole city was able to appreciate it (Figure 10). Also, the artwork was exhibited in Times Square in New York during one month in April 2017, using more than 60 digital billboards there (Figure 11).

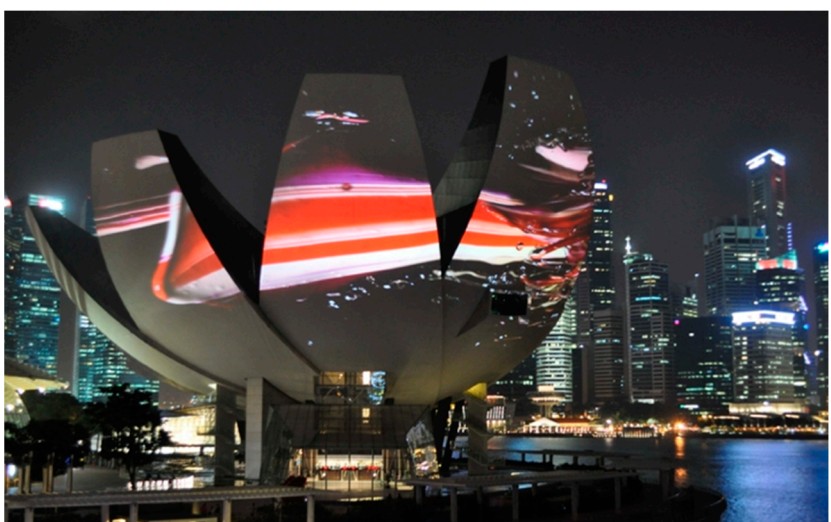

**Figure 10.** Sound of Ikebana projection mapping at ArtScience Museum in Singapore.

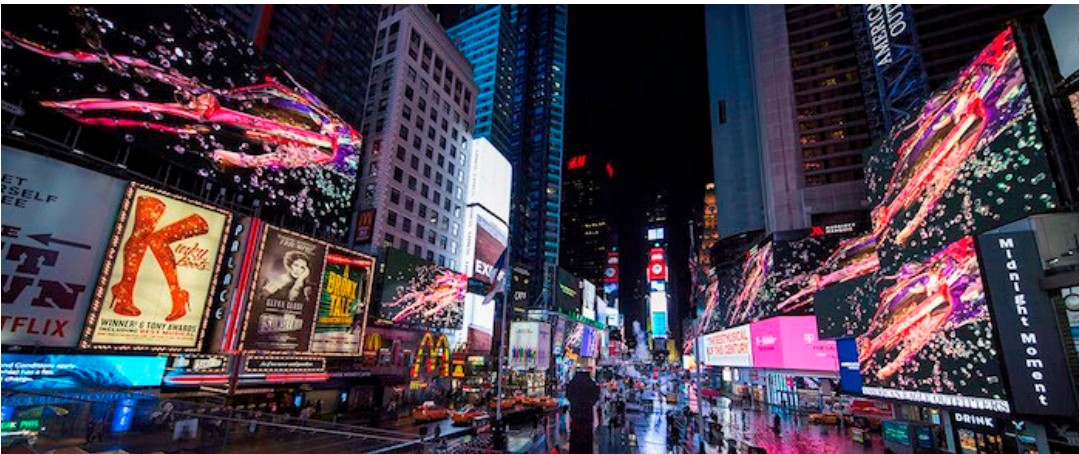

**Figure 11.** Sound of Ikebana exhibited in Times Square, New York.

## 5. Genesis: An Example of Created Art

As another method to create artworks based on fluid dynamics, we have developed a method to let fluid and dry ice bubbles interact to create beautiful forms.

### 5.1. Injection of Paints into Fluid

As a basic material to observe fluid behaviors, we chose color paints. In the work described in the previous section, we chose color paints and succeeded in creating various types of beautiful and mysterious forms by applying vibrations to them (Pang and Tosa 2015; Yunian Pang et al. 2017). Therefore, we are familiar with the behaviors of color paints. This time, instead of giving them sound vibrations, we tried to inject them into water. Based on various preliminary experiments, we found that color paints injected into water from droppers can create interesting forms that resemble the phenomenon of a volcano eruption or hydrothermal vent (Figure 12).

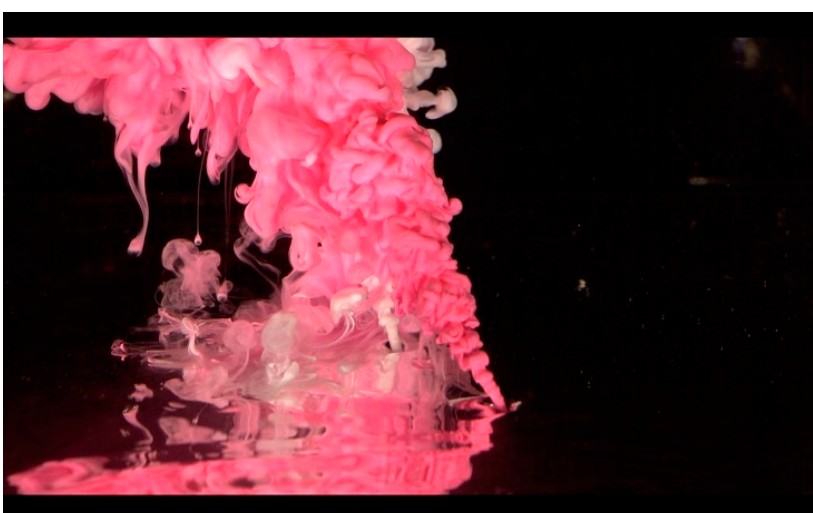

**Figure 12.** Injection of color paints into fluid.

There is some affinity between water and color paints, even in the case of oil-based paints; therefore, injected paints and water mingle rapidly and the water rapidly becomes a kind of "colored water". As what we want to create are the interesting behaviors of injected paints, this rapid mingling process is not preferable. Thus, we tried adding agar into water to increase its viscosity and found that, in the case of water with a certain amount of viscosity, this has the effect of delaying such a mingling process. Additionally, we found that the level of viscosity based on the amount of added agar

plays an important role by changing the mingling time to some extent. This finding was important to create the interesting behavior of injected paints. However, basically behaviors of injected paints are based on the diffusing process and, as the time passes, water and paint are mixed based on a one-way process. Consequently, it is difficult to generate something unexpected based on this basic method. Therefore, some new mechanism of creating unexpected phenomena should be introduced. To realize this, we have introduced the usage of dry ice which is described in the next subsection.

### 5.2. Usage of Dry Ice as Obstacles in Fluid Pathways

Based on fluid dynamics study, we have learned that the existence of obstacles in the pathway of fluid motion is the key to generate beautiful and mysterious forms. At the same time, we have learned that such obstacles should not be fixed ones. Fixed obstacles give fixed effects to the behaviors of fluid and this process is not effective in generating something unexpected. Therefore, such obstacles should move around. Also, it is preferable that the moving patterns of such obstacles are unstable or even unexpected. In addition, it is preferable that forms of the obstacles unexpectedly. We carried out various kinds of experiments to determine such obstacles and finally found that the use of dry ice is very effective as obstacles interacting with injected paints.

Dry ice is the solid form of carbon dioxide. It is used primarily as a cooling agent. Its advantages include a lower temperature than that of water ice and not leaving residue. At the same time, dry ice has been frequently used as a material to create mysterious stage effects, as it creates huge amounts of fog when it is added to water. People have been focusing on the effect of fog generation when they use dry ice. However, we have focused on the early process of fog generation. When dry ice is put into water, based on the temperature difference between water and dry ice, rapid vaporization of dry ice occurs. Many small bubbles, each of which contains carbon dioxide fog, are generated as the result of vaporization and these small bubbles rise from dry ice from bottom to water surface and finally create fog. Watching this process by using a high-speed camera, we have found that such bubbles have interesting forms, with each bubble having a different form. Additionally, during the process of a bubble rising up to the water surface, the bubble always changes its form. This phenomenon gives us the impression that each bubble is a kind of living creature (Figure 13). Then, we had an idea that the combination of these bubbles and injected paints described in the previous subsection would be ideal to generate a new type of phenomenon based on fluid dynamics. Therefore, we have adopted the usage of dry ice as an obstacle material in the pathway of injected fluid.

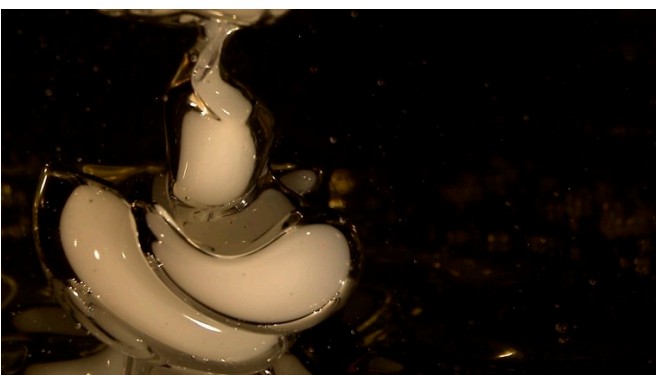

**Figure 13.** Bubbles generated by dry ice.

### 5.3. Genesis: Created Artwork Based on the Interaction between Fluid and Dry Ice

We tried to integrate two methods described in Sections 5.1 and 5.2. Firstly, we put a small block of dry ice into water, letting it generate bubbles with carbon dioxide fog inside. Then a combination of several color paints were injected into the water. Without dry ice-based bubbles, the injected color paints quickly diffused, making the water appear as if colored water. There are two ways to avoid this somewhat uninteresting event. As described in Section 5.1, one way that we have found used agar to

increase water viscosity to some extent. Based on several experiences, we found that there is a certain range of viscosity in which the diffusion of color paints into water occurs slowly. Then, under such a condition, we added dry ice into water. As described in Section 5.2, various dry ice bubbles were generated as the result of the vaporization of dry ice, where forms of dry ice bubbles are different to each other and even their forms changed continuously while rising up in water to the water surface. Then, the injected color paints interacted with these various bubbles and created various complex forms as shown in Figure 14. These created forms were beyond the forms we often see as the result of scientific visualization and look very artistic.

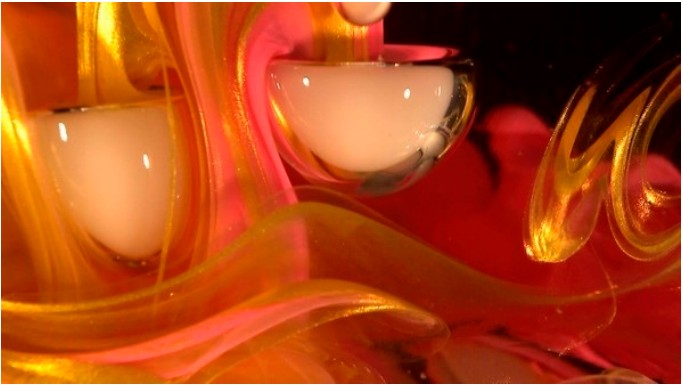

**Figure 14.** An example of the interaction between bubbles created by dry ice and injected color paints.

The created artwork called "Genesis," was exhibited in 12 cities including New York, London, and Paris, and one of the authors, Naoko Tosa, did her world tour as Japan's Cultural Envoy in 2016. One such exhibition carried out in Singapore is shown in Figure 15.

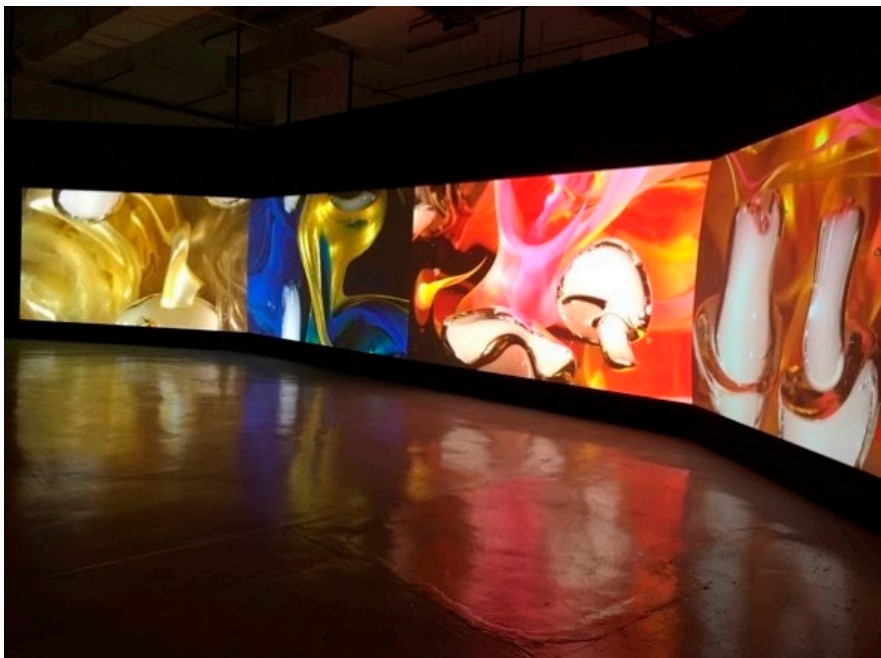

**Figure 15.** Exhibition of "Genesis" at Ikkan Gallery in Singapore in 2017.

## 6. Discussion and Conclusions

In this paper, we proposed new types of media art creation methods and described their details. We have been interested in the art creation process based on the extraction of hidden beauty in nature using technologies. We have noticed and believe that the extraction of hidden beauty is the basis of

Japanese beauty, as one of the authors, Naoko Tosa, received many comments/opinions from people all over the world including art curators and art critics saying that they feel there is Japanese beauty in the artworks developed by her and her team. In Section 2, we discussed this issue by showing several representative Japanese artworks and reached the conclusion that, in Japanese art history, the extraction of hidden beauty in nature and the expression of it as artworks has been the main stream.

We have been interested in fluid behaviors as a natural phenomenon, and have been trying to create artworks by recording fluid behaviors using a high-speed camera. As this area in science is called "fluid dynamics," in Section 3, the explanation of fluid dynamics and also the relationship between fluid dynamics and art were described in detail.

In Sections 4 and 5, two art creation methods based on fluid dynamics were described including their concepts, the methodologies and examples of created artworks. In Section 4, one method of art creation based on fluid dynamics was described. The method is based on the combination of color paints as fluid and sound vibration. We have found that jumping-up color paints, vibrated by sound and shot by a high-speed camera, make beautiful forms and we created an artwork called "Sound of Ikebana" based on the methodology. Both the methodology and the created artwork were described in detail.

In Section 5, another art creation method developed by us, which is based on the combination of two processes, was described. The first method is the effect achieved by injecting color paints into water of various viscosities. The second method is to use dry ice as obstacles that interact with the flow of the injected paints and, based on this, create surprising and mysterious liquid forms. By combining these two methods in a relevant way and also by using a high-speed camera to record and visualize the generated phenomena, we can create beautiful, noble, and inspiring forms.

We think that there are two ways of creating artworks using fluid. One is a purely scientific process and its aim is to find out beauty in the process of liquid motion as a physical phenomenon. In this case, although the created forms look beautiful, the forms do not look artistic, because there is little unexpectedness in the created forms. Another is the usage of liquid as a basic material for creating artwork. Here, the basic process of art creation is controlled by an artist. However, in the final art making process, such as paint throwing and dropping, a randomness, that is one of basic natures of physical phenomena, is included to add value to the created artwork. We have found that our proposed methods situate somewhere between these two different processes. Its feature is that, on one hand, it can keep pure beauty in physical phenomena. On the other hand, our method removes the feeling associated with too scientific phenomena. Therefore, we believe that we have succeeded in creating new type of artworks.

Of course we understand that it is not adequate to connect fluid dynamics directly to artworks including Japanese beauty. We do not want to claim that fluid dynamics-based artworks are the most adequate to express Japanese beauty. At this stage, what we want to claim is the following. We have developed several methods to create beautiful forms based on fluid dynamics. As the creation process is closely related to natural/physical phenomenon and also as Japanese sensitivity has been closely related to beauty included in natural phenomenon, it was easy for one of the authors, a Japanese artist, to include her sensitivity and aesthetics into various art creation processes such as color selection, parameter selection for sound vibration, editing of obtained video and so on. We will further pursue what is Japanese beauty and what is the essential art creation process to include Japanese beauty.

Artificial intelligence (AI) technology is progressing and there are various trials to create artworks using AI (du Sautoy 2019). So far, most of the trials are based on learning existing paintings using deep learning method and creating new paintings that are somewhat similar to the existing paintings. Such paintings are sometimes criticized as they are not new creations. On the other hand, as our art creation methodology is based on physical/natural phenomena, the combination of our methodology and AI would have a chance to create new types of art. In other words, we may have a new type of AI artists in the 21st century.

**Author Contributions:** N.T.—contributed to the creation of artistic concept and also the direction of the art creation. Y.P.—supported N.T. for the creation of "Sound of Ikebana." Q.Y.—supported N.T. for the creation of "Genesis." R.N.—mainly contributed to the direction of the whole project. Also he gave grounding of fluid dynamics based art.

**Funding:** This research received no external funding.

**Conflicts of Interest:** The authors declare no conflict of interest.

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
