# Peer review of "Pursuit and Expression of Japanese Beauty Using Technology"

_arts, 2019_

Reviewer 1 Report

The conceptual creativity of utilizing fluid dynamics to create art is interesting, but it is a little unclear why you consider the created forms "Japanese."

Reviewer 2 Report

This is an interesting and challenging paper to review since the paper makes a number of claims that are rather essentialist and in places not supported by viable argument. However, the work of art discussed is inspiring, interesting and visually very attractive.

The authors’ assumption that they have created a new type of art is premature since they have digitally manipulated data obtained from high speed camera photography of physical experiments where paint is introduced to water infused with dry ice. Their understanding of what Japanese beauty is,  in places is contradictory and can be extended to the way other nations see the relationship between art, nature and natural phenomenon. I do not see in the article is it written now that the authors have created the 'new art' (line 162), either by the failing in the approach they have taken or not being able to convey it in the text.

What makes the works of art as presented in the paper is not that it is a particularly Japanese phenomenon of doing so and by editing the images taken, that could be done by artists from any number of cultures. What makes for me the art works as discussed in the text is: 1) the artistic freedom of doing so; 2) the visual quality of the colours mixed; 3) as well as giving meaning to this work of art, it makes the viewer engage with the wider questions raised by Genesis or Sounds of Ikebana as general concepts within the context of 'fluid dynamics', rather than just being Japanese.

It would be very interesting to publish this article, but I recommend this is only if the authors can further explore and address the issues suggested below, rather than publishing it in the form it is now. The artist’s intentionality that draws on her/his cultural tradition; use of physical phenomenon and new technologies to capture art;  making implicit and explicit artistic choices, e.g. of colours and/or close-up’s of particular ‘happenings’ during the process of paint mixing; giving meaning to works of art ‘as being in the world’ or ‘being part of the world’ where the issues of natural beauty are contested and discussed as part of the global community and in the context of a particular nation. All of these could be usefully explored giving the viewers access to the creative process of art creation,  as well as  affording the artist/s the medium of text in communicating their artistic praxis.

Reviewer 3 Report

The conceptual creativity of utilizing fluid dynamics to create art is inteesting, but it is a little unclear why you consider the created forms "Japanese."   They are also similar to forms from art nouveau and pre-Raphaelite painting, and we could also find parallels in Chinese painting.  Names of pre-modern Japanese artists should probably be in the native format - Katsushika Hokusai, for example, with surname first and pen name following.  Several problems seem like typos - palm for plum, pains for paints,  Also note that Christmas colors are not particularly Japanese.  Art works referenced should be referred to by specific, accurate titles - for example the Hokusai print (NOT a drawing) should include he series title (Fugaku sanjurokkei Kanagawa oki namiura) and be translated (Beneath the Wave off Kanagawa, from 36 Views of Mt. Fuji)  BTW this is a Japanese artist deeply influenced by Western art, so using his work as an example of Japaneseness is a bit suspect. "Wabi-sabi" is a very loose term, not used in a clear way here, and does not define specific colors.

Author Response

Round  2

Reviewer 2 Report

I am happy to accept the changes made by the author/s and recommend the paper for publishing.

Reviewer 3 Report

Much improved over the previous version.  A few inconsistencies, for example, could be cleaned up (for example, "human and nature" should probably be "humans and nature"). The expression of the ideas about what makes Japanese aesthetics is much improved.